# Study to Investigate the Knowledge of Rare Diseases among Dentists, Orthodontists, Periodontists, Oral Surgeons and Craniomaxillofacial Surgeons

**DOI:** 10.3390/ijerph18010139

**Published:** 2020-12-28

**Authors:** Annemarie Kühne, Johannes Kleinheinz, Jochen Jackowski, Jeanette Köppe, Marcel Hanisch

**Affiliations:** 1Department of Cranio-Maxillofacial Surgery, Research Unit Rare Diseases with Orofacial Manifestations (RDOM), University Hospital Münster, D-48149 Münster, Germany; kuehne-am@web.de (A.K.); johannes.kleinheinz@ukmuenster.de (J.K.); 2Department of Oral Surgery and Dental Emergency Care, Faculty of Health, Witten/Herdecke University, Alfred-Herrhausen-Strasse 45, 58455 Witten, Germany; jochen.jackowski@uni-wh.de; 3Institute of Biostatistics and Clinical Research, University of Münster, Schmeddingstraße 56, D-48149 Münster, Germany; jeanette.koeppe@ukmuenster.de

**Keywords:** rare diseases, RDs, knowledge, self-assessment, education, information needs

## Abstract

Fifteen percent of the 5000 to 8000 rare diseases (RDs) can manifest in the oral and maxillofacial region. Little attention has been paid to the care situation of people with RDs in dentistry. Hence, the aim of this study was to assess the level of knowledge about RDs among dentists at a university hospital (DUs) compared to dentists with different professional backgrounds and among general dentists, specialist dentists and DUs in the chamber district of Westfalen-Lippe. Moreover, self-assessment of the level of knowledge was evaluated. A questionnaire was designed, which was made available digitally via a link. A random sample of 1500 dentists, specialist dentists, and oral- and craniomaxillofacial surgeons from the membership of the Dental Association of Westfalen-Lippe, and all dentists, specialist dentists, and oral- and craniomaxillofacial surgeons working at the University Dental Hospitals Münster and Witten/Herdecke, were invited to participate to our study. Differences in the level of knowledge between DUs and non-DUs and differences between DUs, general dentists, and specialist dentists were tested via two-sided Fischer’s exact tests. Differences between the three groups of self-assessment of the level of knowledge and the self-assessment of how sufficient their own knowledge about RDs is were tested via two-sided Kruskal–Wallis tests. The global level of significance was controlled by the Bonferroni method. A total of 267 questionnaires were completed, of which 64.0% were answered by general dentists, 25.5% by specialist dentists and 10.5% by DUs. DUs had a significant higher level of knowledge about RDs (adjusted *p* = 0.012) compared to non-DUs and achieved higher scores (median = 16.5 points) than general (median = 13 points) and specialist dentists (median = 13 points) (*p* = 0.001). In the self-assessments, the differences were not significant (*p* > 0.05). In conclusion, most participants showed no or little knowledge about RDs, and DUs had a significant higher level of knowledge than non-university dentists.

## 1. Introduction

In Germany, approximately 4 million people are affected by a rare disease (RD) [1], and in the EU there are approximately 30 million [2]. However, the meaning of “rare” is subject to temporary and regional variations [3]. “In the European Union (EU), a disease is classified as rare if less than 5 out of 10,000 people are affected” [3]. Fifteen percent of the 5000 to 8000 known RDs can manifest in the dental, oral and maxillofacial area [3]. On top of that, on average, it takes about 8 years before a correct diagnosis can be made [4]. Hence, dentistry is of great importance in the diagnosis and early detection of RDs.

Due to the large number of different RDs, exact knowledge concerning all of them cannot be provided. Therefore, interdisciplinary cooperation between the disciplines of medicine and dentistry is essential [5]. For example, the initial symptoms of (gastrointestinal) Crohn’s disease can occur as oral manifestations, such as ulcerations of the oral mucosa and cheilitis [6]. In ectodermal dysplasia, an abnormality, e.g., conical tooth shape or missing teeth, is part of the symptom triad that characterises this RD [7]. Dentogenic abscesses on caries-free teeth in combination with skeletal alterations may indicate X-linked hypophosphataemia (“phosphate diabetes”) [8]. Recurrent odontogenic keratocysts in combination with multiple basal cell carcinomas can be an indication of Gorlin–Goltz syndrome [9].

As the European Medicines Agency (EMA) reports, despite a continuously increasing number of publications on RDs, only the most common (less than 1000) of the up to 8000 RDs seem to benefit from little medical and scientific knowledge [2].

With the aim of “improving the life situation of every single person with a rare disease”, National Action Alliance for People with Rare Diseases (NAMSE) was founded in 2010 by the Federal Ministry of Health (BMG) together with the Federal Ministry of Education and Research (BMBF) and Alliance of Chronic Rare Diseases (ACHSE) e.V. The focus of the point of care research was to determine what the actual situation of medical and social care of those affected is [10].

The interim reports on the implementation of NAMSE in the Federal Republic of Germany from 2017 and 2019 show that the delayed diagnosis of RDs by physicians and dentists may be due to insufficient knowledge and experience in the treatment of RDs. Therefore, medical and dental education and training should be intensified on the basis of certain proposed measures and solutions in the field of RDs, diagnosis should be accelerated, and strategies for dealing with unclear diagnoses should be developed [11,12]. The aim of this study is to describe the present level of knowledge on RDs among dentists in the chamber district of Westfalen-Lippe.

The idea was inspired by a study among physicians in Belgium, which showed that the average level of knowledge on RDs is “suboptimal” [13]. Therefore, the question arose concerning how much dentists in Germany know about RDs.

Since, up to now, no studies in Germany have examined this topic, a questionnaire was developed to investigate the level of knowledge and self-assessment concerning RDs among dentists. Since this is a study among dentists and oral- and craniomaxillofacial surgeons, the focus is on RDs with orofacial involvement.

## 2. Materials and Methods

### 2.1. Study Design

The study was conducted as an anonymous cross-sectional study among dentists, specialist dentists, and oral- and craniomaxillofacial surgeons in the chamber district of Westfalen-Lippe, where the university dental hospitals of Münster and Witten/Herdecke are located.

The questionnaire see Appendix A was provided digitally; an e-mail with the link to it was sent to the respective practices/clinics. The website Q-Set served as the platform (https://www.q-set.de).

Data collection was carried out between 6 February and 1 May 2020.

### 2.2. The Questionnaire

The questionnaire of the Belgian study served as a guideline for the structure [13]. The questionnaire of the present study was divided into five sections and contained 24 questions (22 single and multiple-choice questions and two visual analogue scales/sliders).

Based on this, four hypotheses (H1 to H4) were tested:

**Hypotheses** **1** **(H1).**
*There is a significant difference in the level of knowledge about RDs between dentists at a university hospital (DUs) and non-university dentists.*


**Hypotheses** **2** **(H2).**
*There is a significant difference in the level of knowledge about RDs between specialist dentists, general dentists and DUs.*


**Hypotheses** **3** **(H3).**
*There is a significant difference in the self-assessment of knowledge about RDs in these three groups.*


**Hypotheses** **4** **(H4).**
*There is a significant difference in these three groups concerning how sufficiently their own knowledge about RDs is assessed.*


The first section of the questionnaire—“general information”—collected general information about the participants, such as gender, age, duration of professional activity, additional (specialist) dental qualifications, dental school and current work location. Therefore, six questions were asked: e.g., “Which university did you study at?”.

In Section 2, Section 3, Section 4 and Section 5, participants were asked about their knowledge about RDs, confrontation/experience with RDs, education/continuing education and information about RDs.

The section “Knowledge about rare diseases” contained one self-assessment and five questions such as “In your opinion a rare disease is…?” or “Which of the following rare diseases, which can manifest themselves orofacially, do you know?”. The RDs queried were those RDs that most frequently affect patients presenting to the special consultation “rare diseases with oral involvement” at the Department of Oral and Cranio-Maxillofacial Surgery at the University Hospital Münster.

The section “Confrontation/Experience with rare diseases” contained one self-assessment and three questions as mentioned below. In the section “Education/Continuing education”, there were four questions, such as “Have you already attended training courses with focus on rare diseases with orofacial manifestations?”. The final section “Information about rare diseases” contained four questions, e.g., “Do you need information concerning rare diseases with orofacial manifestations in your everyday dental practice?”.

Each question was assigned to a specific score. For each correct answer, known RD, source of information or required information and answer “yes”, 1 point was assigned; for each wrong answer, “no”, “none”, “no information” or omitted question, 0 points were awarded. The evaluation of the answers could thus be based on the total number of points achieved for the respective section and could be classified according to a division into categories (see Appendix A).

The sections “Knowledge about rare diseases” and “Confrontation/Experience with rare diseases” were the basis for the examination of the primary objective, “Knowledge on rare diseases”. Both began with a self-assessment evaluating knowledge about RDs (Section 2) and the adequacy of this knowledge (Section 3). Then, a knowledge survey on knowledge about RDs was carried out using exam-type questions, and the study aimed to find out whether dentists, specialist dentists, and oral- and craniomaxillofacial surgeons have (clinical) experience with RDs. Therefore, the third section, “Confrontation/Experience with rare diseases”, was to determine whether the participant had already treated a patient affected by a RD, has thought during the treatment that there might be a RD, or has ever diagnosed a RD.

The evaluation of the level of knowledge was based on a division into four categories. The target score could have values of 1 = “no knowledge” (0–10 points), 2 = “little knowledge” (11–16 points), 3 = “good knowledge” (17–22 points) and 4 = “very good knowledge” (23–28 points).

The self-assessments of knowledge about RDs could be determined by analogue scales/sliders (section “Knowledge about rare diseases”) and show whether the knowledge was perceived as sufficient (section “Confrontation/Experience with rare diseases”). The scales could take values between 1 and 10, whereas the value of 1 corresponded to “very good” (section “Knowledge about rare diseases”) or “yes, I know enough about RDs” (section “Confrontation/Experience with rare diseases”). The value of 10 corresponded to “inadequate” (section “Knowledge about rare diseases”) or “I do not know enough about RDs at all” (section “Confrontation/Experience with rare diseases”). In this case, the participant could not see the exact numerical value of the scale because it was not visibly scaled, so that the most intuitive answer was possible. Only the extreme values 1 and 10 were labeled and were visible to the participants for orientation. For the question “How do you assess your knowledge about rare diseases?”, these were defined as “very good” (on the left) and “inadequate” (on the right). For the question “Do you think your knowledge about rare diseases is sufficient?” they were “yes” (on the left) and “not at all” (on the right).

Second, under the heading “Education/Continuing Education”, it was first investigated whether RDs were addressed during dental education or whether training courses were attended later and whether the participants know, where they can obtain relevant information and where they obtain their knowledge. Here, the categorisation was “no/little” (0–3 points), “moderate” (4–6 points) and “(very)good education/continuing education” (7–10 points).

Then, it was determined whether there is a need for information about RDs or not (section “Information about rare diseases”). This was divided into the categories “no” (0–5 points), “little” (6–10 points) and “considerable information needs on rare diseases” (11–14 points).

### 2.3. Ethics and Sample Size Calculation

The Ethics Commission of the Medical Association of Westfalen-Lippe and the Westphalian Wilhelms University gave a positive ethics vote for the implementation of our study (ref. No. 2019-672-f-S). A total of four primary hypotheses were tested, and the local significance levels were therefore adjusted using the Bonferroni method to consider the multiple test problem. The global level of significance was a priori set to 5%. A closed testing procedure was planned to test the hypotheses H3 and H4. Using a two-sided Kruskal–Wallis test, first the difference in the distributions in all three groups were tested to a local significance level of α = 1.25%. If the respective null hypothesis could be rejected, all pairwise comparisons using a two-sided Mann–Whitney-U test were performed at the same significance level of α = 1.25% [14]. The sample size was calculated in such a way that a mean difference of 1.0 (standard deviation 2.0) for the self-assessments can be shown with a power of 80% and a local level of significance of α = 1.25%, resulting in *n* = 259 evaluable participants. Currently, there is insufficient information on the expected effect sizes available from the published literature. Thus, the calculation was done as a worst-case scenario using the smallest relevant difference in content. Regarding the hypotheses H1 and H2, using a two-sided chi-square test, respectively, an effect of w² = 0.1 can be proven with a power of 80% and a local significance level of α = 1.25% for both hypotheses. A sample size of *n* = 259 evaluable participants (*n* = 288 participants, considering a dropout rate of 10%) is therefore sufficient for all four hypotheses.

### 2.4. Participants

All members of the Dental Association of Westfalen-Lippe were searched out, counted, and numbered according to the specialties of oral surgery, craniomaxillofacial surgery and orthodontics, as well as all general dentists. A random sample of 1500 dentists, specialist dentists, and oral- and craniomaxillofacial surgeons from the membership of the Dental Association of Westfalen-Lippe, and all dentists, specialist dentists, and oral- and craniomaxillofacial surgeons working at the University Dental Hospitals Münster and Witten/Herdecke were invited to participate to our study.

The sample was drawn in such a way that the proportion of specialist and general dentists as well as oral- and craniomaxillofacial surgeons approximately corresponded to the original distribution within the chamber (2019 of a total of approximately 6525 active members [15]). Thus, 1295 general dentists and 205 specialist dentists, among them 98 orthodontists, 66 oral surgeons, and 41 oral- and craniomaxillofacial surgeons, were invited to participate in the study.

At the time of the survey, 82 dentists were working at the University Dental Hospital Münster and 56 dentists were employed in Witten/Herdecke.

Under the umbrella term “specialist dentist”, the possible additional qualifications of oral surgery, oral- and craniomaxillofacial surgery, orthodontics and/or “other” (free answer possibility) could be selected. For the evaluation of the results, all individual answers were examined. It was decided to include the additional qualification for periodontology among the specialist dentists. Thus, in this study, only those additional qualifications were used for defining the term “specialist dentist”, which define the term “specialist dentist” according to the ZÄKWL and KZVWL (Zahnärztekammer und Kassenzahnärztliche Vereinigung Westfalen-Lippe) [16].

For DUs, no distinction was made between general and specialist dentists in the evaluation. The current work location university hospital classified this group of participants.

### 2.5. Statistical Analysis

The statistical analysis was performed using IBM SPSS Statistics for Windows, Version 26.0., IBM Corp., Armonk, NY, USA and SAS software V9.4, SAS Institute Inc., Cary, NC, USA. Differences in the knowledge about RDs between DUs and non-DUs (H1) were tested via two-sided Fischer’s exact tests, since the conditions for the two-sided chi-square test were not fulfilled. Differences in the levels of knowledge (H2) and other categorical variables between the three groups—DUs, general dentists and specialist dentists—were tested via two-sided chi-square tests or two-sided Fischer’s exact tests, if the expected frequencies were too small. If the exact test could not be performed because of invalid computational time, an asymptotic approximation via the Monte Carlo method was done to estimate the respective test statistic.

For continuous variables, such as self-assessments (verification of H3 and H4) and the scores obtained for the respective sections, a single-factor ANOVA using a two-sided Kruskal–Wallis test was performed.

The *p*-values for the four primary hypotheses were adjusted using the Bonferroni method and compared to a global significant level of 5%. All other *p*-values were not adjusted and are purely explorative.

## 3. Results

### 3.1. General Information

A total of 267 dentists answered the questionnaire in full, including 104 (39.0%) women and 163 (61.0%) men (see Table 1). The distribution in the age groups showed that 9% were under 30 years of age, 43.1% were between 30 and 50 years old, and 48% were over 50 years of age. Ninety-eight participants did not complete the questionnaire, so their answers were not considered in the following analysis. Among the 267 participants, 171 were general dentists, 68 were specialist dentists (239 non-university dentists) and 28 were DUs. Among the specialist dentists, a total of 28 oral surgeons, 9 oral and craniomaxillofacial surgeons, and 19 orthodontists participated (see Table 1).

The response rate was 13.2% among general dentists, 33.2% among specialist dentists, and 20.3% among DUs.

### 3.2. Results of the Sections “Knowledge about Rare Diseases” and “Confrontation/Experience with Rare Diseases”

It is noticeable that for the question “In your opinion, a rare disease is…”, the options of answer 3 and 4 were chosen most frequently in all three groups. Overall, 64.3% of DUs, 57.4% of specialist, and 48% of general dentists estimated that a RD is a disease affecting no more than five in 10,000 people in the EU, which was the correct answer. About one third of DUs (32.1%) and specialist dentists (35.3%) and 43.3% of general dentists estimated that no more than five in 250,000 people in the EU are affected.

In total, 39.3% of DUs knew that 15% of RDs manifest themselves in the craniomaxillofacial region. This differed from the responses of the other two groups (*p* = 0.029); here, 17.6% of the specialist and 15.8% of general dentists answered correctly.

When asked “Which statements on rare diseases do you think are correct?”, there was a statistically striking difference (*p* = 0.002) in the three groups: DUs and general dentists answered more statements correctly than specialist dentists (*p* = 0.001, *p* = 0.013). For example, more DUs (85.7%) than general (59.6%) and specialist dentists (47.1%) chose the answer option “The majority of rare diseases are genetically co-related” (*p* = 0.002). Similarly, a larger proportion of DUs (39.3%) chose the answer option “Rare diseases manifest themselves primarily in early childhood” than general (21.1%) and specialist dentists (10.3%) (*p* = 0.005). Furthermore, more DUs (39.3%) than general (19.3%) and specialist dentists (7.4%) (*p* = 0.001) knew that “In the European Union, approximately 30 million people suffer from a rare disease”.

When asked which RDs are known to the participants, there was a statistically striking difference (*p* = 0.012) in the answers of the three groups (some examples are shown in Figure 1): DUs knew more RDs than specialist (*p* = 0.007) and general dentists (*p* = 0.004), 75.0% knew eight or more RDs, and none of the DUs knew fewer than three RDs.

Among general and specialist dentists, the number of known RDs hardly differed (*p* = 0.916, both median = 8); for DUs, the median was 11 (IQR = 5.75).

In estimating the time span after which a RD is diagnosed, the majority of DUs (64.3%) and only about one-fifth of specialist (22.1%) and one-quarter of general dentists (24.6%) estimated “after more than three years” (*p* = 0.003).

The first question in the section “Confrontation/Experience with rare diseases” “Have you ever treated a patient affected by a rare disease/have you ever seen such a patient before?” was answered in the affirmative by more DUs (85.7%) and specialist- (82.4%) than general dentists (69.0%) (*p* = 0.037).

Over 70% of the respective group had already thought that a RD may be present when treating a patient (70.2% of the general dentists, 73.5% of the specialist dentists, and 75.0% of the DUs).

For the question “Have you ever diagnosed a rare disease?” there were striking differences in the choice of the answer options “yes, once”, “yes, several times”, “no, never” and “no information” (*p* < 0.001). More DUs (53.6%) answered “yes, several times” than specialist (36.8%) and general dentists (16.4%) did. Overall, 10.3% of specialist and about a quarter of general dentists (24.6%) had diagnosed a RD once.

More than half of the general dentists (53.8%), 41.2% of the specialist dentists and 35.7% of the DUs had never diagnosed a RD (in total, almost half of all participants, 48.7%).

### 3.3. Knowledge about RDs

In the sections “Knowledge about rare diseases” and “Confrontation/Experience with rare diseases”, the level of knowledge and experience regarding RDs were assessed and divided into the respective categories according to the number of points achieved. A graphical representation of the scores of all sections of the questionnaire can be found in Appendix A. It is shown that DUs had a significantly better level of knowledge about RDs than dentists in a different current work location (adjusted *p* * < 0.001) (H1) (see Figure 2A). There was also a significant difference in the level of knowledge between the three groups (adjusted *p* * = 0.012) (H2) (see Table 2 and Figure 2B).

For the achieved scores, there was a statistically noticeable difference between general- and specialist dentists and DUs (*p* = 0.001) (see Table 2).

### 3.4. Self-Assessments

In the self-assessments concerning the level of knowledge about RDs, there was no significant difference (adjusted *p* * = 0.196) between the three groups: the knowledge was assessed neither as very good nor as inadequate (see Table 2 and Figure 3).

In the section “Confrontation/Experience with rare diseases”, general, specialist dentists and DUs all indicated that they assess their knowledge as rather insufficient, although here again there was no significant difference between the three groups (median in all three groups = 7, adjusted *p* * > 0.05) (see Table 2 and Figure 3).

### 3.5. Education/Continuing Education about RDs

The majority of participants in the three groups (43.3% of general dentists, 47.1% of specialist dentists and 46.4% of DUs) indicated that during their dental education, “too little time” was spent on acquiring knowledge about RDs, their diagnostics, and therapy.

Approximately one fifth of DUs (21.4%) reported that no time was spent on RDs, and approximately one fifth of general dentists (21.1%) said that sufficient time was spent on them; 20 participants (7.5%) indicated that a lack of knowledge about RDs was due to the lack of training time on RDs, including more general dentists (7.6%) and specialist dentists (8.8%) than DUs (3.6%).

There was a statistically noticeable difference in the information on continuing education in the three groups (*p* = 0.006): 53.6% of DUs, 48.5% of specialist dentists and only 28.1% of general dentists had already attended training courses with focus on RDs. More than half of the general dentists (55.6%), a third of the specialist dentists (33.8%), and 28.6% of the DUs had not trained but would like to do so (47.2% in total). A total of 10.5% of participants were interested in taking part in further training.

In all three groups, about three quarters of the participants said they knew where to get information on RDs when treating a patient or person with a RD. Approximately one fifth of general dentists (21.6%) and one-quarter of DUs said they did not know.

As a source of knowledge on RDs, the studies served clearly more general dentists (55.6%) than specialist dentists (39.7%) and DUs (39.3%) (*p* = 0.043). More specialist (61.8%) than general dentists (43.9%) and DUs (46.4%) obtained their knowledge from subject-specific online portals (*p* = 0.043). More DUs (75%) than specialist (54.4%) and general dentists (43.9%) obtained their knowledge from colleagues (*p* = 0.006).

### 3.6. Information Needs Concerning the RDs

Considerably more general dentists (14.6%) did not know where to get information about RDs (*p* = 0.031); in comparison, only three specialist dentists and one DU did not know where to get information (see Table 3).

Information on incidence and prevalence was needed by about half of the general dentists (49.1%); for the specialist dentists, it was 47.1%, and for the DUs, it was about 39.3%. Information on lethality and mortality was needed by about one third of DUs (32.1%) and specialist dentists (35.3%), and 26.9% of general dentists. Regarding treatment modalities, more than 85% of the participants in the respective group (in total 88.4% of participants) required information, and between 59 and 70% (in total 61% of participants) required information about relevant medications.

When asked about organizations, websites and sources of information about RDs, the majority of the general (70.2%) and specialist dentists (66.2%), but only just under a third of the DUs (28.6%), did not know any of those mentioned in the questionnaire. More DUs knew the websites ROMSE e.V. (*p* < 0.001) and Orphanet (*p* = 0.002) and the organizations NAMSE (*p* = 0.005) and ACHSE e.V. (*p* = 0.001) than in the case of general and specialist dentists.

In the concluding question, only one DU considered knowledge on RDs to be absolutely unimportant (*p* > 0.05), and two general dentists stated that RDs virtually play no role at all in everyday dental practice (*p* > 0.05) (see Table 4).

## 4. Discussion

Currently, no studies in Germany have examined the level of knowledge about RDs among dentists, specialist dentists and oral- and craniomaxillofacial surgeons. Therefore, the aim of this study was to uncover potential gaps in the knowledge concerning RDs and possible areas of improvement in terms of the lack of dental education/continuing education.

The primary hypothesis (H1) has been confirmed: DUs perform significantly better in comparison to non-university dentists, as shown by numerous examples in the results. 

There were also striking differences in the three groups—general dentist, specialist dentist and DU—which likewise confirmed the second hypothesis (H2). This correlates with a survey among GPs in Bulgaria, which showed that their knowledge and awareness of RDs is low [17]. In Belgium, however, it was found that GPs have less knowledge on RDs than specialists [13]. The different distribution could be due to the smaller number of cases in this study.

H3 and H4 have not been verified. In the three groups, there was no significant difference in the self-assessments of knowledge about RDs (H3) and how sufficient it was perceived (H4); general dentists showed a tendency towards the direction of inadequate. This could be due to their less experience with RDs, as they indicated in the third section. Furthermore, all participants showed a tendency to perceive their knowledge about RDs as rather insufficient, which correlated with the mostly low level of knowledge about RDs.

In the USA, GPs (general practitioners) also rated their knowledge about RDs lower (“fair or poor”) compared to specialists. The majority of GPs rated their own knowledge about RDs as neutral or ineffective (more than specialists) [18].

As the results show, three quarters of the participants had no or little knowledge about RDs; only 3% had very good knowledge. This reflects the situation that, for example, up to seven to eight years can pass between the first occurrence and the diagnosis of a RD [4].

When diagnosing some RDs, such as Marfan syndrome or Ehlers–Danlos syndrome [19], even more time passed [20], and the suffering of those affected is, therefore, understandable [21].

Little knowledge of the possible symptoms of a RD, such as Ehlers–Danlos syndrome, can lead to increased bleeding or insufficient local anaesthesia, presenting the dental surgeon with unexpected problems. [4]

Often, the dentist is the first who can make the correct diagnosis or even a suspected diagnosis of a RD and, therefore, could refer the patient to a specialist [22].

Earlier study results show that there was also a need for education, training, and continuing education in dentistry when dealing with other complex clinical pictures: a survey among dentists showed that basic dental ethical knowledge was available in the dental profession, but uncertainties existed in more complex situations, such as the treatment of HIV-positive patients [23]. According to the Fédération Dentaire Internationale (FDI), the dentist is often the one who can diagnose the first oral manifestations of AIDS [24,25].

Clinical experience with RDs was evident for all participants but more frequently among DUs and specialist dentists. According to the EurordisCare2 and 3 surveys, in Germany, it was mainly specialised centres, ”hospital consultations” or private practices where RDs were diagnosed [26]. In addition, in a study from the USA, patients indicated that the doctor who diagnosed the RD was usually a local specialist [18].

Although these results are based on surveys without a dental background, they support the finding that the majority of DUs and 36.8% of specialist dentists stated that they had diagnosed an RD several times, that no DU had diagnosed an RD only once, and that over half of the general dentists had never diagnosed a RD.

The fact that, altogether, nearly half of all participants had never diagnosed a RD could be due to the fact that the survey was limited to the chamber district of Westfalen-Lippe and that two centres for the treatment of RDs are located in this area: CeSER (Centre for Rare Diseases Ruhr) and CSE Münster (Centre for Rare Diseases Münster).

The majority of participants reported that during dental education, too little time was spent on acquiring knowledge on RDs, their diagnostics and therapy. Therefore, targeted seminars could be a way to create awareness of RDs and their manifestations.

At the University of Lille, for example, there was a workshop for medical students to improve their RD awareness and knowledge on RDs so that in certain treatment situations, they could keep a suspected diagnosis of a RD in mind [27]. The expert interviews of Vandeborne et al. also showed that physicians should be educated about RDs during their studies. In addition, both GPs and specialists should be sensitised to RDs, and teaching methods should start with patient characteristics [13].

Overall, RDs have received little attention in dentistry to date [28]. Those dentists who participated in this study, however, were interested in further education in the field of RDs through continuing education. This implies that there is a willingness, to deal with RDs in the future. The awareness that a RD may be present and that, in the case of a suspected diagnosis, interdisciplinary cooperation is not shied away from would be a great benefit for those affected and could take away the uncertainty about their symptoms. A survey in the USA showed that patients also believed that physicians should seek help in the process of finding a diagnosis and refer patients to specialists [18].

More than half of the participants stated that they needed information on RDs. However, not all of them knew where to obtain it. The fact that healthcare professionals (HCPs) and GPs did not know where information about RDs can be obtained has already been shown in previous studies among medical professionals [13]. Therefore, field case reports in dental journals, for example about RDs in everyday dental care, could be helpful. Information about institutions, centres, practices, or colleagues to whom the dentist can turn could provide guidance in dealing with affected patients.

Congresses at which patients affected by RDs report on their experience and life with the disease could also bring the topic closer to the dental profession. In 2017, the first national congress on RDs in dental, oral and craniomaxillofacial medicine was held in Münster, Germany, where dentists and physicians as well as patients affected by RDs spoke and exchanged their experiences regarding RDs. A large number of participants reflected their lively interest in RDs [28].

Since nearly 20% of the participants unfortunately did not have time for research, the thematization of RDs during the studies, for example by integrating them into the curriculum, would be a possibility to create a “knowledge background” to RDs.

The fact that significantly more DUs were familiar with the websites and information sources mentioned in the questionnaire correlated with their higher level of knowledge about and experience with RDs. Earlier studies showed that only a few of those affected and only a few GPs knew organisations or sources of information about RDs [13,29].

The fact that the greatest need for information was about treatment modalities and relevant medications and that the majority of participants considered knowledge about RDs and their differential diagnostic significance important (general dentists > specialist dentists > DUs) correlated with the fact that the majority of participants had already seen patients affected by a RD.

The greater experience of DUs with RDs may have caused their consideration of knowledge about RDs as more important than specialist- and general dentists did.

Until now, the care situation of people with RDs has not been a focus of dentistry; few data are available on the oral health-related quality of life of those affected [28]. However, these data describe a negative correlation between oral health-related quality of life and the occurrence of oral symptoms in RDs [4]. Therefore, it is essential for the dentist to have knowledge about RDs in order to carry out prevention and early detection and to give attention to people affected by RDs by using patient-centred diagnostics [13] and treatment.

### 4.1. Outlook of the Study and Future Perspectives

For a further assessment of the results of this study, it would be interesting to compare the results with a larger cohort and, consequently, a larger number of cases, for example by conducting a Germany-wide study. To exclude regional differences, the comparison with other chamber districts could also be informative.

In our opinion, society should aim to achieve a higher quality of dental care for people with RDs in the future. Therefore, targeted information should be disseminated on RDs that may manifest in the dentist’s specialty. This can be done, for example, through topic-related training courses or publications in relevant journals.

### 4.2. Study Limitations

The results of this study are based purely on the answers of the study participants. The self-assessments showed how they perceived their knowledge individually. It is not possible to show what prompted them to do so and whether this correlates with the treatment situation of their patients.

Furthermore, these are the first results of a survey of the state of knowledge about RDs among dentists. Therefore, unfortunately, only a comparison with the knowledge of physicians and their handling of the diagnostics and therapy of RDs can be made. This can only be reflected concerning the background of studies in the field of human medical education, further education, and training [13,17,18].

## 5. Conclusions

The majority of participants considered knowledge about RDs to be very important in everyday dental practice.

General dentists as well as specialist ones mostly had no or little knowledge about RDs, whereas DUs had a much higher level of knowledge. The self-assessments showed that all participants assessed their knowledge as neither very good nor inadequate but showed a tendency towards insufficient knowledge.

Therefore, it can be concluded that the current knowledge of the dental profession regarding RDs does not seem to be sufficient for adequate early detection and therapy.

This article is intended as an indication of how much more knowledge is needed in dentistry to help provide people with RDs a shorter path from the onset of symptoms to diagnosis and possible treatment.

## Figures and Tables

**Figure 1 ijerph-18-00139-f001:**
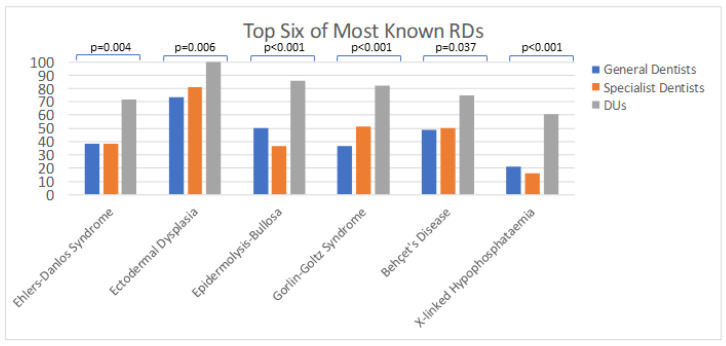
Data in %. Differences between the groups of dentists were tested via two-sided chi-square-test. Rare disease (RD).

**Figure 2 ijerph-18-00139-f002:**
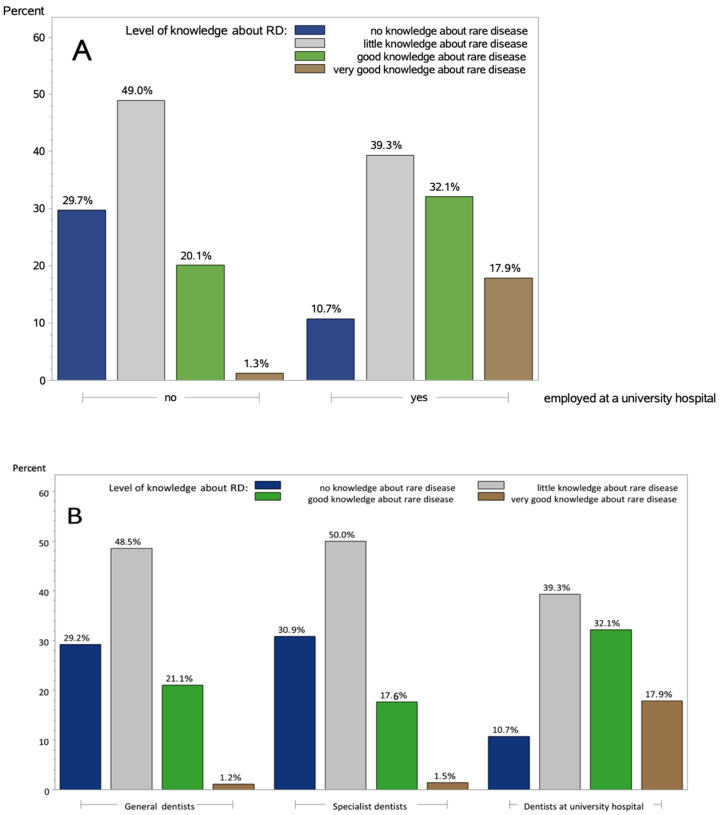
Level of knowledge. (**A**) Comparing dentists employed at vs. outside a university hospital. There were significant differences in the level of knowledge between dentists at a university hospital and dentists in a different current work location (tested via two-sided exact Fisher test, *p* * < 0.001 adjusted using the Bonferroni method). (**B**) Comparing general dentists, specialist dentists and dentists at a university hospital. There were significant differences in the level of knowledge between general dentists, specialist dentists and dentists at a university hospital (tested via two-sided exact Fisher test, *p* * = 0.012 adjusted using the Bonferroni method).

**Figure 3 ijerph-18-00139-f003:**
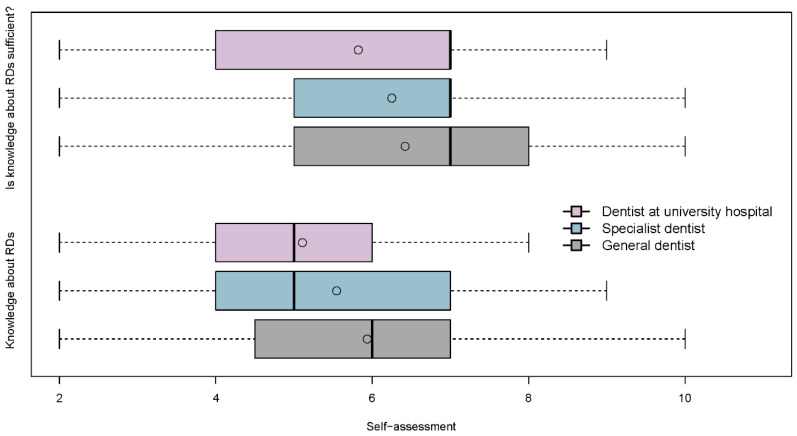
Self-assessment of knowledge about rare diseases (RDs) and self-assessment whether their own knowledge about RDs is sufficient. Evaluated on a scale with direction from positive (left) to negative (right) being 1 = very good knowledge/yes, knowledge is sufficient and 10 = inadequate knowledge/knowledge is not sufficient at all. Participants could only see the labels of the limits 1 and 10. Differences between the groups of dentists were tested via two-sided Kruskal–Wallis tests and adjusted using the Bonferroni method. For both self-assessments, no significant differences between the professional groups were observed (adjusted *p*-value > 0.05).

**Table 1 ijerph-18-00139-t001:** General information about the participants.

General Information	All Participants	General Dentists	Specialist Dentists	Dentists at University Hospital (DUs)	*p*-Value ^1^
Frequency entire chamber district of Westfalen-Lippe—*n* (%)	6663 (100.0%)	5637 (84.6%)	888 (13.3%)	138 (2.1%)	-
Frequency—*n* (%)	267(100.0%)	171 (64.0%)	68 (25.5%)	28 (10.5%)	-
Female sex—*n* (%)	104 (39.0%)	68 (39.8%)	24 (35.3%)	12 (42.9%)	0.737
Age—*n* (%)					0.001
<30 years	24 (9.0%)	21 (12.3%)	0 (0.0%)	3 (10.7%)	
30–40 years	62 (23.2%)	31 (18.1%)	12 (17.6%)	19 (67.9%)	
41–50 years	53 (19.9%)	24 (14.0%)	25 (36.8%)	4 (14.3%)	
51–60 years	79 (29.6%)	59 (34.5%)	19 (27.9%)	1 (3.6%)	
>60 years	49 (18.4%)	36 (21.1%)	12 (17.6%)	1 (3.6%)	
Professional experience—*n* (%)					<0.001
<5 years	34 (12.7%)	27 (15.8%)	0 (0.0%)	7 (25.0%)	
5–10 years	41 (15.4%)	19 (11.1%)	8 (11.8%)	14 (50.0%)	
11–15 years	20 (7.5%)	10 (5.8%)	6 (8.8%)	4 (14.3%)	
16–20 years	28 (10.5%)	17 (9.9%)	10 (14.7%)	1 (3.6%)	
>20 years	144 (53.9%)	98 (57.3%)	44 (64.7%)	2 (7.1%)	
Subject area:—*n* (%)					
General dentist	181 (67.8%)	171 (100.0%)	0 (0.0%)	10 (35.7%)	
Oral surgery	28 (10.5%)	0 (0.0%)	17 (25.0%)	11 (39.3%)	
Oral- and craniomaxillofacial surgery	9 (3.4%)	0 (0.0%)	4 (5.9%)	5 (17.9%)	
Orthodontics	19 (7.1%)	0 (0.0%)	18 (26.5%)	1 (3.6%)	
Current work location—*n* (%)					-
University hospital	28 (10.5%)	0 (0.0%)	0 (0.0%)	28 (100.0%)	
Private hospital	6 (2.2%)	4 (2.3%)	2 (2.9%)	0 (0.0%)	
self-employed in a medical practice	154 (57.7%)	107 (62.6%)	47 (69.1%)	0 (0.0%)	
Joint practice	61 (22.8%)	44 (25.7%)	17 (25.0%)	0 (0.0%)	
Medical care center	10 (3.7%)	8 (4.7%)	2 (2.9%)	0 (0.0%)	
Others	8 (3.0%)	8 (4.7%)	0 (0.0%)	0 (0.0%)	

^1^ Differences between groups of dentists for categorical variables were tested using two-sided chi-square tests or two-sided Fischer’s exact tests, if the expected frequencies were <5.

**Table 2 ijerph-18-00139-t002:** Results of Section 2, Section 3, Section 4 and Section 5
^2^ of the questionnaire. IQR = interquartile range.

	All Participants	General Dentists	Specialist Dentists	Dentists at University Hospital (DUs)	*p*-Value
**Results of the questionnaire**
Self-assessment of knowledge about rare disease—median (IQR)	6(7 − 4 = 3)	6(7 − 4 = 3)	5(7 − 4 = 3)	5(6 − 4 = 2)	0.196 *
Number of points Section 2—median (IQR)	12(14 − 8 = 6)	11(14 − 8 = 6)	10(13 − 8 = 5)	15(18 − 11.25 = 6.75)	<0.001
Self-assessment how adequate knowledge about rare diseases is—median (IQR)	7(8 − 5 = 3)	7(8 − 5 = 3)	7(7 − 5 = 2)	7(7 − 4 = 3)	1.000 *
Total score of Knowledge about rare diseases—median (IQR)	13(16 − 10 = 6)	13(16 − 10 = 6)	13(16 − 10 = 6)	16.5(21 − 14 = 7)	0.001
Level of knowledge about rare disease—*n* (%)					0.012 *
no knowledge	74 (27.7%)	50 (29.2%)	21 (30.9%)	3 (10.7%)
little knowledge	128 (47.9%)	83 (48.5%)	34 (50.0%)	11 (39.3%)
good knowledge	57 (21.3%)	36 (21.1%)	12 (17.6%)	9 (32.1%)
very good knowledge	8 (3.0%)	2 (1.2%)	1 (1.5%)	5 (17.9%)
Number of points Section 4—median (IQR)	4(6 – 3 = 3)	4(5 – 3 = 2)	5(6 – 3 = 3)	5(6 – 3 = 3)	0.146
Education/continuing education about RDs—*n* (%)					0.062
no/little education	37 (13.9%)	21 (12.3%)	11 (16.2%)	5 (17.9%)
moderate education	106 (39.7%)	79 (46.2%)	20 (29.4%)	7 (25.0%)
(very) good education	124 (46.4%)	71 (41.5%)	37 (54.4%)	16 (57.1%)
Number of points Section 5—median (IQR)	5(6 – 4 = 2)	5(6 – 4 = 2)	5(6 – 4 = 2)	6(7 – 5 = 2)	0.024
Information needs concerning the RDs—*n* (%)					0.384
no information needs	146 (54.7%)	97 (56.7%)	38 (55.9%)	11 (39.3%)
little information needs	118 (44.2%)	72 (42.1%)	29 (42.6%)	17 (60.7%)
considerable information needs	3 (1.1%)	2 (1.2%)	1 (1.5%)	0 (0.0%)

^2^Section 2: “Knowledge about rare diseases”, Section 3: “Confrontation/Experience with rare diseases”, Section 4: “Education/Continuing Education”, Section 5: “Information about rare diseases”. Differences between groups of dentists for categorical variables were tested using the two-sided chi-square test or two-sided Fischer’s exact test, if the expected frequencies were <5. Group differences regarding continuous variables were tested using a two-sided Kruskal–Wallis test. * *p*-values of the primary hypotheses were adjusted using the Bonferroni method; all other p-values were unadjusted and fully explorative.

**Table 3 ijerph-18-00139-t003:** Do you need information concerning rare diseases with orofacial manifestations in your everyday dental practice?

	All Participants—*n* (%)	General Dentists—*n* (%)	Specialist Dentists—*n* (%)	Dentists at University Hospital (DUs)—*n* (%)
Yes.	144 (53.9%)	85 (49.7%)	42 (61.8%)	17 (60.7%)
Yes, but I do not know where to get this information.	29 (10.9%)	25 (14.6%)	3 (4.4%)	1 (3.6%)
Yes, but unfortunately I do not have time for research.	53 (19.9%)	41 (24.0%)	7 (10.3%)	5 (17.9%)
No, I am sufficiently informed.	21 (7.9%)	11 (6.4%)	6 (8.8%)	4 (14.3%)
No, because I am not interested…	6 (2.2%)	2 (1.2%)	3 (4.4%)	1 (3.6%)
No information	22 (8.2%)	14 (8.2%)	7 (10.3%)	1 (3.6%)

Participants could choose several answers.

**Table 4 ijerph-18-00139-t004:** Do you, as a dentist, consider it important to have knowledge about RDs that manifest orofacially?

	All Participants—*n* (%)	General Dentists—*n* (%)	Specialist Dentists—*n* (%)	Dentists at University Hospital (DUs)—*n* (%)
Yes. I consider it to be very important.	134 (50.2%)	80 (46.8%)	37 (54.4%)	17 (60.7%)
Yes, knowledge about RDs has an important differential diagnostic significance.	153 (57.3%)	102 (59.6%)	37 (54.4%)	14 (50.0%)
One should have heard about RDs.	77 (28.8%)	54 (31.6%)	18 (26.5%)	5 (17.9%)
No, it is unimportant.	1 (0.4%)	0 (0.0%)	0 (0.0%)	1 (3.6%)
No, RDs virtually play no role at all in everyday dental practice.	2 (0.7%)	2 (1.2%)	0 (0.0%)	0 (0.0%)

Participants could choose several answers. Rare disease (RD).

## Data Availability

The data presented in this study are available on request from the corresponding author. The data are not publicly available due to privacy.

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
