# Peer review of "Study to Investigate the Knowledge of Rare Diseases among Dentists, Orthodontists, Periodontists, Oral Surgeons and Craniomaxillofacial Surgeons"

_ijerph, 2020, doi:10.3390/ijerph18010139_

Round 1

Reviewer 1 Report

The article addresses important topics for dentistry and rare diseases. However, the referred manuscript needs to go through some adjustments to make its publication possible.

Attached file.

Author Response

We would like to thank the editor and the reviewers for their time spent on reviewing our manuscript and their helpful comments. Their suggestions have been implemented in the manuscript. In this letter, we respond point-by-point to the comments and explain the revisions.

All changes to the manuscript were highlighted using the "Track Changes" function in Microsoft Word.

We hope the manuscript is now suitable for publication in the International Journal of Environmental Research and Public Health.

Reviewer 1:

The article addresses important topics for dentistry and rare diseases. However, the referred manuscript needs to go through some adjustments to make its publication possible.

Title

-       Ok.

Abstract

  • I suggest better describing the location of the study. “A random sample of 1500 dentists, specialist dentists and oral- and craniomaxillofacial surgeons from the membership of the Dental Association of Westfalen-Lippe and all dentists, specialist dentists and oral- and craniomaxillofacial surgeons working at the University Dental Hospitals Münster and Witten/Herdecke were invited to participate to our study.” (page 1 lines 35-39)
  •  
  • Answer: Thank you for noticing. Information about participants was added:
  • The authors report in the “Only little attention is paid to the care situation of people with RDs in dentistry. Furthermore, there is a negative correlation between the occurrence of oral manifestations and the oral health-related quality of life of those affected”, but they do not consider this topic in a considerable way in the introduction of the study. “Furthermore, there is a negative correlation between the occurrence of oral manifestations and the oral health-related quality of life of those affected” as it does not fully fit the objective of this study and have revised the introduction section. (page 1 lines 30-31)
  •  
  • Answer: We have deleted this sentence:

Introdution

  • Very superficial.
  • Authors do not present a considerable justification for carrying out the study. Answer: Thank you very much for this note. We added an explanation:
  •  

“15% of the 5000 to 8000 known RDs can manifest in the dental, oral and maxillofacial area. On top of

that, on average it takes about 8 years before a correct diagnosis can be made [4]. Hence, dentistry is of

great importance in the diagnosis and early detection of RDs.“ (page 2 lines 59-62)

  • The importance of conducting studies on this topic is rarely addressed in the text. of rare diseases. One of the reasons for this is the lack of knowledge about rare diseases knowledge of rare diseases among dentists through this study. We have added two “With the aim of "improving the life situation of every single person with a rare disease", NAMSE of Health (BMG) together with the Federal Ministry of Education and Research (BMBF) and ACHSE social care of those affected is [10]. (page 2 lines 77-87)
  •  
  • The interim reports on the implementation of NAMSE in the Federal Republic of Germany from 2017 and 2019 show that the delayed diagnosis of RDs by physicians and dentists may be due to insufficient knowledge and experience in the treatment of RDs. Therefore, medical and dental education and training should be intensified on the basis of certain proposed measures and solutions in the field of RDs, diagnosis should be accelerated, and strategies for dealing with unclear diagnoses should be developed.
  • e.V. The focus under the point of care research was to determine what the actual situation of medical and
  • (National Action Alliance for People with Rare Diseases) was founded in 2010 by the Federal Ministry
  • paragraphs to this effect in the introduction section:
  • among physicians and dentists. In this respect, it was our aim to gain insight into the
  • Answer: In the meantime, politics has recognized the problems in the diagnosis and therapy
  • Authors should better address the relevance of the study. of rare diseases. One of the reasons for this is the lack of knowledge about rare diseases knowledge of rare diseases among dentists through this study. We have added a paragraph “The interim reports on the implementation of NAMSE in the Federal Republic of Germany from 2017 and 2019 show that the delayed diagnosis of RDs by physicians and dentists may be due to insufficient knowledge and experience in the treatment of RDs. Therefore, medical and dental education and training should be intensified on the basis of certain proposed measures and solutions in the field of RDs, diagnosis should be accelerated, and strategies for dealing with unclear diagnoses should be developed [11,12].” 
  • (page 2 lines 82-87)
  • to this effect in the introduction section:
  • among physicians and dentists. In this respect, it was our aim to gain insight into the
  • Answer: In the meantime, politics has recognized the problems in the diagnosis and therapy
  • The last paragraph would be more appropriate in the topic methods.
  • Answer: The paragraph has been added to “Methods”. Thank you very much for this note. (page 3 lines 118-126)

Methods

  • Best term: cross-sectional study.  
  • Answer: Thank you for this note. The term has been changed.
  • I suggest a better description of the place of origin of the sample. We added this information to the methods section (page 5 lines 203-209). 
  •  
  • Answer: “All members of the Dental Association of Westfalen-Lippe were searched out, counted and numbered according to the specialties of oral surgery, craniomaxillofacial surgery and orthodontics, as well as all general dentists. A random sample of 1500 dentists, specialist dentists and oral- and craniomaxillofacial surgeons from the membership of the Dental Association of Westfalen-Lippe and all dentists, specialist dentists and oral- and craniomaxillofacial surgeons working at the University Dental Hospitals Münster and Witten/Herdecke were invited to participate to our study.”

Results

  • Extensive.  Answer: Since the scores and self-assessments were not normally distributed, we have calculated the median and associated IQR instead. If variables were normally distributed, the median would also be valid in this case, since the median and mean would then coincide.
  • - Add mean and standard deviation of study participants.
  • Answer: We shortened the results.

- The authors repeat the table values in the result section. Table is not cited in the text.

Answer: Thank you for noticing. Citations have been added in the text.

  • I suggest that “Table 1” be divided into two: one for general information, another for the results of the questionnaire.  
  • Answer: We divided “Table 1” into two: one for general information, another for the results of the questionnaire.
  • Tables improperly formatted. 
  • Answer: The table has been reformatted. We hope that all tables now meet the requirements of the journal.

Discussion

  • Authors repeat the results found in the study.  
  • Answer: Thank you very much for noticing. We have now revised the discussion and avoided repeating the results.
  • Add a paragraph discussing the relevance of the topic studied. “(…) up to seven to eight years can pass between the first occurrence and the diagnosis of a RD [19]. When diagnosing some RDs, such as the Marfan syndrome or the Ehlers-Danlos syndrome, even more time passed [21], and the suffering of those affected is, therefore, understandable [22].“The awareness that a RD may be present and that in the case of a suspected diagnosis interdisciplinary cooperation is not shied away from would be a great benefit for those affected and could take away the uncertainty about their symptoms.” (page 17 lines 525-528) 
  • “More than half of the participants stated that they needed information on RDs. However, not all of them knew where to obtain it. The fact that HCPs (healthcare professionals) and GPs did not know where information about RDs can be obtained has already been shown in previous studies among medical professionals. Therefore, field case reports in dental journals, for example about RDs in everyday dental care, could be helpful. Information about institutions, centres, practices or colleagues to whom the dentist can turn could provide guidance in dealing with affected patients.” (page 17 lines 530-536)
  • (…) Often the dentist is the first who can make the correct diagnosis or even a suspected diagnosis of a RD and, therefore, could refer the patient to a specialist [23].” (page 16 lines 479-485)
  • Answer: We added a paragraph discussing the relevance of the topic studied:
  • The authors could reflect on how individuals with Rare Diseases are affected by the little knowledge of dental surgeons.
  • Answer: We added this paragraph in the discussion section:

“Little knowledge of the possible symptoms of a rare disease, such as Ehlers-Danlos Syndromes, can lead

to increased bleeding or insufficient local anesthesia, presenting the dental surgeon with unexpected

problems [19].” (page 16 lines 483-485)

Conclusions - Ok.

Reviewer 2 Report

The manuscript is well written.  However, verbs are presented in both present and paste tense.  This should be changed to be consistent, preferably using past tense.  Also, page 3, lines 124-131, verb "should" should be replaced with was, or similar.

Methods should include a more complete description of questionnaire and individual questions. If that was done, results could be presented more concisely.  Results for individual questions could be presented in a table for ease. 

Instead of referring to your sections by numbers, it would be clearer to refer to them by a descriptive name - so that there is no need for the reader to remember or refer back to the numbers. 

The statement, "All three groups assessed their knowledge as neither very good nor inadequate, but rather as insufficient (median =7)" is confusing.  It uses a double negative.  Rather than describe what they didn't report, could you describe what they did report?  

You report median scores with no reference to what they indicate.  What is the range.  Is 7 a good median score?

Page 2- line 68 "only little"  is confusing here.

Were specialists other than the three listed  (ortho, oral surgeons) excluded?  What about pedodontists?  periodonists? 

page , line 90- use of parentheses is confusing, avoid this. 

Page 2, lines 69-70 - statement seems irrelevant.   

The questionnaire description is unclear.  Name the sections rather than number them.  More clearly state at the beginning that there were exam type questions used to assess knowledge followed by self assessment of the adequacy of this knowledge.  This distinction is lost a bit here.  If you do not present all of the questions - present at least an example of a question for  each section, and the number of questions each section includes. Most readers will not read the suppliment.   Present results and conclusions in the same order.  

Is there a case when { specialist is also a DU?  HOw is that person categorized?

Page 3, line 101 "place of study" is this dental school, or city?  Please be specific, similarly, "professional environment"  does this mean current work location?  

Table 1 should be divided into 2 tables, one for general info on the subjects and another for Results. Do not describe methods in in the title of the table.  Again, in your table, what do the Means indicate?  In your methods, clarify how overall scores were calculated for the graphs that appear in figure one.

Figure 2 needs more labeling.  Which direction is positive on the sliding scale?  Some of your descriptions of this scale imply it is categorical "neither very good nor inadequate?  How were these categories defined?  

How were the "known RDs" determined?  The results would be clearer if individual question results were presented before the combined results instead of vice versa.  Again, perhaps using a table rather than the large volume of text. 

Your discussion is primarily a restatement of your results, this should be avoided. You have a heading for study limitations-but the paragraph that follows doesn't include only limitations.  Your conclusion is weak. 

page 13, line 444, "human physicians?"  THis entire paragraph is unclear.

Author Response

We would like to thank the editor and the reviewers for their time spent on reviewing our manuscript and their helpful comments. Their suggestions have been implemented in the manuscript. In this letter, we respond point-by-point to the comments and explain the revisions.

All changes to the manuscript were highlighted using the "Track Changes" function in Microsoft Word.

We hope the manuscript is now suitable for publication in the International Journal of Environmental Research and Public Health.

Reviewer 2:

The manuscript is well written.  However, verbs are presented in both present and paste tense.  This should be changed to be consistent, preferably using past tense. 

Answer: The verbs have been changed using past tense. We hope the tenses now meet the requirements of the journal.

Also, page 3, lines 124-131, verb "should" should be replaced with was, or similar.

Answer: The term has been changed.

Methods should include a more complete description of questionnaire and individual questions. If that was done, results could be presented more concisely.  Results for individual questions could be presented in a table for ease. 

Answer: Thank you for noticing. In the methods section “2.2. The questionnaire” was rewritten according to your recommendations.

Instead of referring to your sections by numbers, it would be clearer to refer to them by a descriptive name - so that there is no need for the reader to remember or refer back to the numbers. 

Answer: Numbers of sections have been replaced by their headings.

The statement, "All three groups assessed their knowledge as neither very good nor inadequate, but rather as insufficient (median =7)" is confusing.  It uses a double negative.  Rather than describe what they didn't report, could you describe what they did report?  

Answer: See next comment below.

You report median scores with no reference to what they indicate.  What is the range.  Is 7 a good median score?

Answer: A median score of 7 shows a tendency towards insufficient knowledge (in the section “Confrontation/Experience with rare diseases”). The statement of the individual indication of the participants on the scale can only show a tendency towards the direction of the extreme values (1= yes, knowledge is sufficient, 10=knowledge is not sufficient at all).

It implies that the participants may have heard about RDs but that they have no idea how to deal with such a patient.

The ranges showed no significant differences between the groups. Therefore, they are only mentioned in the table (see Table 2).

Page 2- line 68 "only little" is confusing here.

Answer: The term “only” has been deleted.

Were specialists other than the three listed (ortho, oral surgeons) excluded?  What about pedodontists?  periodonists? 

Answer: Thank you very much for this question. We added an explanation:

Under the umbrella term “specialist dentist”, the possible additional qualifications of oral surgery, oral- and craniomaxillofacial surgery, orthodontics and/or “other” (free answer possibility) could be selected. For the evaluation of the results, all individual answers were examined. It has been decided to include the additional qualification for periodontology among the specialist dentists. Thus, in this study, only those additional qualifications were used for the definition of the term specialist dentist, which define the term "specialist dentist" according to the ZÄKWL and KZVWL (Zahnärztekammer und Kassenzahnärztliche Vereinigung Westfalen-Lippe) [16] ” (page 5 lines 217-223)

page , line 90- use of parentheses is confusing, avoid this. 

Answer: Thank you for this note. The parentheses have been removed.

Page 2, lines 69-70 - statement seems irrelevant.   

Answer: We have removed this statement, because it does not fully fit into the topic of the introduction.

The questionnaire description is unclear. 

Answer: The methods section “2.2. The questionnaire” was rewritten according to your recommendations, see above.

Name the sections rather than number them. 

Answer: The sections are now named by their headings.

More clearly state at the beginning that there were exam type questions used to assess knowledge followed by self-assessment of the adequacy of this knowledge.  This distinction is lost a bit here. 

Answer: Thank you very much for this note. This note has been implemented. (line)

If you do not present all of the questions - present at least an example of a question for  each section, and the number of questions each section includes. Most readers will not read the suppliment.   

Answer: This proposal has been implemented.

Present results and conclusions in the same order.  

Answer: Thank you very much for noticing. The results are now in the same order in all sections, always taking into account the order of our working hypotheses.

Is there a case when { specialist is also a DU?  HOw is that person categorized?

Answer: We added an explanation.

“For DUs, no distinction was made between general and specialist dentists in the evaluation. The current work location university hospital classified this group of participants.” (page lines 224-225)

Page 3, line 101 "place of study" is this dental school, or city?  Please be specific, similarly, "professional environment"  does this mean current work location?  

Answer: The questions were: “Which university did you study at?” and “Most of your time you work at…?”.

So the terms were changed into the suggested ones: “place of study” into “dental school” and “professional environment” into “current work location”.

Thank you very much for this linguistic note.

Table 1 should be divided into 2 tables, one for general info on the subjects and another for Results.

Answer: Table 1 has been divided into 2 tables, see above.

Do not describe methods in in the title of the table. 

Answer: The description of methods has been deleted.

Again, in your table, what do the Means indicate?  In your methods, clarify how overall scores were calculated for the graphs that appear in figure one.

Answer: We have added an explanation regarding the means.

Since the scores and self-assessments were not normally distributed, we have calculated the median and associated IQR instead. If variables were normally distributed, the median would also be valid in this case, since the median and mean would then coincide. (see above)

There was a data error in Figure 1 which has been fixed. We added an explanation regarding score-calculation:

“The section “Knowledge about rare diseases” contained one self-assessment and five questions such as “In your opinion a rare disease is…?” or “Which of the following rare diseases, which can manifest themselves orofacially, do you know?”. The RDs queried were those RDs that most frequently affect patients presenting to the special consultation “rare diseases with oral involvement” at the Department of Oral and Cranio-Maxillofacial Surgery at the University Hospital Münster.”     (page 3 lines 133-137)

“Each question was assigned to a specific score. For each correct answer, known RD, source of information or required information and answer “yes”, 1 point was assigned; for each wrong answer, “no”, “none”, “no information” or omitted question, 0 points were awarded. The evaluation of the answers could thus be based on the total number of points achieved for the respective section and could be classified according to a division into categories (see Table 2 and Supplement S2).” (pages 3-4 lines 144-147)

“The evaluation of the level of knowledge was based on a division into four categories. The target score could have values of 1=“no knowledge” (0–10 points), 2=“little knowledge” (11–16 points), 3=”good knowledge” (17–22 points) and 4=”very good knowledge” (23–28 points).” (page 4 lines 159-161)

Figure 2 needs more labeling.  Which direction is positive on the sliding scale? 

Answer: Labeling is completed within explanation in the footer/caption.

“Self-assessment of knowledge about RDs and self-assessment whether their own knowledge about RDs is sufficient. Evaluated on a scale with direction from positive (left) to negative (right) being 1=very good knowledge/yes, knowledge is sufficient. 10=inadequate knowledge/knowledge is not sufficient at all. Participants could only see these labels of the limits [1,10]. (…)” (page 12-13 Figure 3, lines 344-348. It is now named Figure 3 because individual question results are now presented before the combined results as you suggested.)

Some of your descriptions of this scale imply it is categorical "neither very good nor inadequate?  How were these categories defined?  

Answer: The participants were able to intuitively indicate on the scale how they assess their individual level of knowledge:

“(…) it was not visibly scaled” for the participant, “so that the most intuitive answer was possible.” (page 4 lines 170-171)

A scale of 1 to 10 was available for the evaluation, so that the result could assume a constant value. “Only the extreme values 1 and 10 were labeled and were visible to the participants for orientation” (page 4 lines 171-172) (for the section "Knowledge about rare diseases": 1=very good, 10=inadequate, for the section "Confrontation/Experience with rare diseases"; 1=yes, knowledge is sufficient, 10=knowledge is not sufficient at all).

How were the "known RDs" determined? 

Answer: The RDs queried have been determined as follows:

They are those RDs that most frequently affect patients presenting to the special consultation "rare diseases with oral involvement" at the Department of Oral and Cranio-Maxillofacial Surgery at the University Hospital Münster. (page 3, lines 135-137)

The results would be clearer if individual question results were presented before the combined results instead of vice versa.  Again, perhaps using a table rather than the large volume of text.

Answer: Thank you very much for this note. We exchanged results section 3.4. “Results of the sections Knowledge about rare diseases and Confrontation/Experience with rare diseases” with results section 3.2. “Knowledge about rare diseases” and 3.3. “Self-Assessments.”

Your discussion is primarily a restatement of your results, this should be avoided.

Answer: Some of the results have been removed from the discussion.

But the results that we believe to be relevant to the discussion have been left in. We hope this is okay.

You have a heading for study limitations-but the paragraph that follows doesn't include only limitations. 

Answer: The “study limitations” have been revised so that only the limitations of the study are listed here. (page 18 lines 574-582)

The remaining aspects are now listed under the new heading "Outlook of the study and Future perspectives". (page 18 lines 564-572)

One point has been added in the discussion to avoid the restatement of our results:

“Overall, RDs have received little attention in dentistry to date [28]. Those dentists who participated in this study, however, were interested in further education in the field of RDs through continuing education. This implies that there is a willingness, to deal with RDs in the future.” (page 17 lines 521-525)

Your conclusion is weak:

Answer: This was added:

“Therefore, it can be concluded that the current knowledge of the dental profession regarding RDs does not seem to be sufficient for adequate early detection and therapy. This article is intended as an indication of how much more knowledge is needed in dentistry to help provide people with RDs a shorter path from the onset of symptoms to diagnosis and possible treatment.” (page 19 lines 604-608)

page 13, line 444, "human physicians?"  THis entire paragraph is unclear.

Answer: The term “human” has been deleted. The sentence has been revised and is now divided into two sentences. (page 18 line 578-582)

Round 2

Reviewer 1 Report

The authors made significant improvements to the revised version and addressed reviewer concerns appropriately.  I recommend this manuscript for acceptance.